# Differentially Private Linear Sketches: Efficient Implementations and Applications

**Fuheng Zhao**[*][†]
fuheng_zhao@ucsb.edu

**Dan Qiao**[*][†]
danqiao@ucsb.edu

**Rachel Redberg**[*]
rredberg@ucsb.edu

**Divyakant Agrawal**[*]
agrawal@cs.ucsb.edu

**Amr El Abbadi**[*]
amr@cs.ucsb.edu

**Yu-Xiang Wang**[*]
yuxiangw@ucsb.edu

## Abstract

Linear sketches have been widely adopted to process fast data streams, and they can be used to accurately answer frequency estimation, approximate top K items, and summarize data distributions. When data are sensitive, it is desirable to provide privacy guarantees for linear sketches to preserve private information while delivering useful results with theoretical bounds. We show that linear sketches can ensure privacy and maintain their unique properties with a small amount of noise added at initialization. From the differentially private linear sketches, we showcase that the state-of-the-art quantile sketch in the turnstile model can also be private and maintain high performance. Experiments further demonstrate that our proposed differentially private sketches are quantitatively and qualitatively similar to noise-free sketches with high utilization on synthetic and real datasets.

## 1 Introduction

Data sketches are fundamental tools for data analysis, statistics, and machine learning [Cormode and Yi, 2020]. Two of the most widely studied problems in data summaries are frequency estimation and quantile approximation. Many real world applications need to estimate the frequency of each item in the database and understand the overall distribution of the database. These applications include stream processing [Das et al., 2009, Bailis et al., 2017], database management [Misra and Gries, 1982, Metwally et al., 2005, Zhao et al., 2022], caching [Zakhary et al., 2020], system monitoring [Gupta et al., 2016, Ivkin et al., 2019, Zhao et al., 2021], federated learning [Rothchild et al., 2020], among others.

On one hand, the motivation for data sketch algorithms is to efficiently process a large database and extract useful knowledge, since computing the exact information for a large amount of data is both time and memory intensive. For instance, Munro and Paterson [1980] proved that to find the true median of a database with $n$ items using $p$ sequential passes requires at least $\Omega(n^{1/p})$ memory. On the other hand, to protect user-level privacy, privacy-preserving algorithms limit the disclosure of private information in the database so that an observer cannot infer much about an individual. Recent works have shown that data sketches can be integrated with privacy-enhancing technologies to provide insightful information and preserve individual privacy at the same time [Cormode, 2022].

Differential privacy [Dwork et al., 2006] is a widely-accepted definition of privacy. Recently, researchers have observed that some data sketches are inherently differentially private [Blocki et al., 2012, Smith et al., 2020], while many other data sketches need modifications to the algorithm to be

---

[*]Department of Computer Science, UC Santa Barbara.
[†]The first two authors contributed equally.

36th Conference on Neural Information Processing Systems (NeurIPS 2022).

differentially private. In particular, a substantial amount of literature has focused on differentially private data sketches for tasks such as linear algebra [Upadhyay, 2014, Arora et al., 2018], cardinality estimation [Mir et al., 2011, Pagh and Stausholm, 2021, Dickens et al., 2022] and quantile approximation [Tzamos et al., 2020, Gillenwater et al., 2021, Alabi et al., 2022].

In this paper, we introduce new differentially private algorithms that support both insertions and deletions for frequency, top k, and quantile approximation. While many data sketches assume an insertion-only model [Greenwald and Khanna, 2001, Shrivastava et al., 2004, Karnin et al., 2016] or a bounded-deletion model [Jayaram and Woodruff, 2018, Zhao et al., 2022, 2021], our algorithms build on top of linear sketches [Charikar et al., 2002, Cormode and Muthukrishnan, 2005] and operate in the turnstile model, which allows an arbitrary number of insertions and deletions into the database. Earlier, researchers attempted to prove CountSketch [Charikar et al., 2002] itself preserves differential privacy, but the authors acknowledged that there are issues in the proof [Li et al., 2019]. Instead of proving that linear sketches, i.e., both Count-Min and CountSketch, are inherently differentially private, we add a small amount of Gaussian noise at their initialization to provide a privacy guarantee, while maintaining linear sketches' original properties, providing high utility for frequency and top K estimations, and keeping update and query algorithms unchanged. We also demonstrate that our analysis provides the tight uniform bound (Section 3.1 and Appendix E). In addition, we propose the first differentially private quantile sketch in the turnstile model by leveraging the differentially private linear sketch. Our differentially private sketches can be queried an arbitrary number of times without affecting privacy guarantees based on the post-processing immunity. Following prior works [Choi et al., 2020, Smith et al., 2020], we assume ideal random hash functions exist, and this assumption can be replaced in practice by cryptographic hash functions [Dickens et al., 2022].

## 2 Preliminaries

Consider a database $X = \{i_t\}_{t \in [N]}$ of $N$ items that are drawn from a large *universe* of size $U$, such as IPv4 address of size $2^{32}$, and for each insert or delete operations, one item can be inserted into or deleted from the database $X$. To support ordered statistic such as quantile, we assume that the *universe* is some finite totally ordered data universe.

**Definition 2.1.** *Given a database $X$, the frequency of an item $x$ is $f(x) = \sum_{t=1}^{N} \pi(i_t = x)$ where $\pi$ returns 1 if $i_t$ is $x$, and 0 otherwise.*

**Definition 2.2.** *Given a database $X$ of items drawn from an ordered universe, the rank of an item $x$ is $R(x) = \sum_{t=1}^{N} \pi(i_t \leq x)$ where $\pi$ returns 1 if $i_t$ is less or equal to $x$ and 0 otherwise.*

Given the large size of $N$, calculating the actual statistics, such as frequency and quantile, is often hard, and hence most applications are satisfied with an *approximation*. The *randomized frequency estimation problem* takes an accuracy parameter $\gamma$ and a failure probability $\beta$ such that, for any item $x$, $|\widehat{f}(x) - f(x)| \leq \gamma \cdot N$ with high probability $1 - \beta$, where $\widehat{f}(x)$ is the estimated frequency and $f(x)$ is the true frequency [Cormode and Hadjieleftheriou, 2008]. In addition, the *randomized quantile approximation problem* also takes an accuracy parameter $\gamma$ and a failure probability $\beta$ such that, for any item $x$, $|\widehat{R}(x) - R(x)| \leq \gamma \cdot N$ with high probability $1 - \beta$ where $\widehat{R}(x)$ is the estimated rank and $R(x)$ is the actual rank [Karnin et al., 2016].

### 2.1 Differential Privacy

**Definition 2.3.** *Databases $X$ and $X'$ are neighbors ($X \sim X'$), if they differ in at most one element.*

Through this paper, we assume the *update/replace* definition of differential privacy instead of *add/remove* definition of differential privacy, in which one item in $X$ is updated or replaced by another item in $X'$ [Vadhan, 2017].

**Definition 2.4** (Differential Privacy [Dwork et al., 2006])**.** *A randomized algorithm $M$ satisfies $(\epsilon, \delta)$-differential privacy ($(\epsilon, \delta)$-DP) if for all neighboring databases $X$, $X'$ and for all possible events $E$ in the output range of $M$, we have*

$$\mathbb{P}(M(X) \in E) \leq e^{\epsilon} \cdot \mathbb{P}(M(X') \in E) + \delta.$$

When $\delta = 0$, $\epsilon$-DP is known as pure DP, and when $\delta > 0$, $(\epsilon, \delta)$-DP is known as approximate DP.

**Definition 2.5** (Gaussian Mechanism [Dwork et al., 2006]). *Define the $\ell_2$ sensitivity of a function $f : \mathbb{N}^{\mathcal{X}} \mapsto \mathbb{R}^d$ as*

$$\Delta_2(f) = \sup_{neighboring\ X, X'} \|f(X) - f(X')\|_2.$$

*The Gaussian mechanism $\mathcal{M}$ with noise level $\sigma$ is then given by*

$$\mathcal{M}(X) = f(X) + \mathcal{N}(0, \sigma^2 I_d).$$

Specifically, the Gaussian mechanism is known to satisfy a stronger notion of privacy known as zero-concentrated differential privacy (zCDP, defined below); zCDP lies between pure and approximate DP and can be translated into standard DP notations, as shown in Lemma 2.9. Moreover, zCDP satisfies cleaner composition theorems, as shown in Lemma 2.7.

**Definition 2.6** (zCDP [Dwork and Rothblum, 2016, Bun and Steinke, 2016]). *A randomized mechanism $M$ satisfies $\rho$-Zero-Concentrated Differential Privacy ($\rho$-zCDP), if for all neighboring databases $X, X'$ and all $\alpha \in (1, \infty)$,*

$$D_\alpha(M(X)\|M(X')) \le \rho\alpha,$$

*where $D_\alpha$ is the Renyi divergence [Van Erven and Harremos, 2014].*

**Lemma 2.7** (Adaptive composition and Post Processing of zCDP [Bun and Steinke, 2016]). *Let $M : \mathcal{X}^n \to \mathcal{Y}$ and $M' : \mathcal{X}^n \times \mathcal{Y} \to \mathcal{Z}$. Suppose $M$ satisfies $\rho$-zCDP and $M'$ satisfies $\rho'$-zCDP (as a function of its first argument). Define $M'' : \mathcal{X}^n \to \mathcal{Z}$ by $M''(x) = M'(x, M(x))$. Then $M''$ satisfies $(\rho + \rho')$-zCDP.*

**Lemma 2.8** (Privacy Guarantee of Gaussian mechanism [Dwork et al., 2014, Bun and Steinke, 2016]). *Let $f : \mathbb{N}^{\mathcal{X}} \mapsto \mathbb{R}^d$ be an arbitrary d-dimensional function with $\ell_2$ sensitivity $\Delta_2 = \sup_{neighboring\ X, X'} \|f(X) - f(X')\|_2$. Then for any $\rho > 0$, Gaussian Mechanism with parameter $\sigma^2 = \frac{\Delta_2^2}{2\rho}$ satisfies $\rho$-zCDP.*

**Lemma 2.9** (Converting zCDP to DP [Bun and Steinke, 2016]). *If $M$ satisfies $\rho$-zCDP then $M$ satisfies $(\rho + 2\sqrt{\rho \log(1/\delta)}, \delta)$-DP.*

As we use exclusively Gaussian mechanisms and their composition in our proposed algorithms, our method actually satisfies $(\epsilon, \delta)$-DP guarantees with stronger parameters than what is implied by zCDP via techniques from [Balle and Wang, 2018, Dong et al., 2019], which reduces the $\epsilon$ parameter by a sizable fraction in typical parameter regimes. We stick to zCDP for clarity and generality, because all our results would apply without changes if we modify the noise into other mechanisms satisfying zCDP, e.g., the Discrete Gaussian Mechanism [Canonne et al., 2020].

## 2.2 Revisiting Linear Sketches

Charikar et al. [2002] proposed the **CountSketch**, a randomized algorithm that summarizes a database and solves the frequency estimation problem. The CountSketch uses a $d \times w$ array of counters, i.e, C[d, w], where all the counters are initialized to **zero**, and has two sets of independent hash functions $h$ and $g$. For each row $r$, the hash function $h_r$ maps input items uniformly onto $\{1, \ldots, w\}$ and the hash function $g_r$ maps input items uniformly onto $\{-1, +1\}$. For item $x$ with value $v \in \{-1, +1\}$, CountSketch updates $d$ counters, one per each row, based on the hash values such that for a particular row $r$, $g_r(x)$ will be added or subtracted to the counter at the $h_r(x)^{th}$ index depending on whether $x$ is being inserted or deleted respectively, as shown in Algorithm 1. Hence, the update time is $O(d)$. To estimate the frequency of item $x$, CountSketch will output the $\text{median}_{1 \le r \le d}\ g_r(x) \cdot C[r, h_r(x)]$, as shown in Algorithm 2. By updating each row's counter based on the hashed value of either 1 or $-1$ and reporting the median for query, CountSketch provides an unbiased estimate. To reduce the failure probability of bad estimations, $d$ is set to $O(\log(1/\beta))$ and it uses $O(\frac{1}{\gamma} \log(\frac{1}{\beta}))$ space to solve the frequency estimation problem.

---

**Algorithm 1** Linear Sketch Update$(x, v)$

---

1: **Input:** Item $x$ with value $v \in \{-1, +1\}$, counter arrays $C$, and two sets of hash functions $\{h_1, \ldots, h_{C.rows}\}$ and $\{g_1, \ldots, g_{C.rows}\}$.
2: **for** $r \leftarrow 1, 2, \ldots, C.rows$ **do**
3: $\quad C[r, h_r(x)] \leftarrow C[r, h_r(x)] + v \cdot g_r(x)$
4: **end for**
5: **Output:** $C$.

---

Cormode and Muthukrishnan [2005] proposed the **Count-Min** sketch that shares the same initialization, update, and data structure as CountSketch. Count-Min sketch also uses $O(\frac{1}{\gamma}\log(\frac{1}{\beta}))$ space to solve the frequency estimation problem. A major difference is that Count-Min sketch makes all hash functions in set $g$ return positive 1. As a result, to estimate the frequency of item $x$, Count-Min sketch returns $\min_{1\leq r\leq d} C[r, h_r(x)]$ instead of the median, as shown in Algorithm 2. In addition, it has the nice property of never underestimating item's frequency. Since linear sketches can approximate an item's frequency accurately, they also solves the top K approximation problem by returning the K items associated with the highest estimated frequency.

---
**Algorithm 2** Linear Sketch Query$(x)$
---
1: **Input:** Item $x$, counter arrays $C$, and two sets of hash functions $\{h_1, \ldots, h_{C.rows}\}$ and $\{g_1, \ldots, g_{C.rows}\}$.
2: arr $\leftarrow [\,]$
3: **for** $r \leftarrow 1, 2, \ldots, C.rows$ **do**
4:     arr.append$(g_r(x) \cdot C[r, h_r(x)])$
5: **end for**
6: **Output:** $\min(\text{arr})$ **for Count-Min or** $\text{median}(\text{arr})$ **for CountSketch.**

---

Gilbert et al. [2002] made the connection between frequency and quantiles, in which the quantile range can be decomposed into at most $\log U$ dyadic intervals [Cormode et al., 2019] and the sum of the estimated frequencies for these intervals gives the estimated rank. Wang et al. [2013] leveraged the unbiased property of CountSketch and proposed the Dyadic CountSketch (DCS) to estimate the frequencies of each dyadic interval. For more specific details, Appendix B and [Cormode and Yi, 2020] provide a comprehensive analysis of quantile sketches.

## 3 Private Linear Sketches

In this section, we present new algorithms for differentially private linear sketches. We highlight that our Private Count-Min and CountSketch only require a different initialization while they share the same update (Algorithm 1) and query (Algorithm 2) with the original Count-Min and CountSketch. Therefore, the implementation of our algorithms is efficient. Below we show our private initialization.

---
**Algorithm 3** DP Linear Sketch Initialization with Gaussian Noise
---
1: **Input:** Desired accuracy parameter $\gamma$, failure probability $\beta$, and budget for zCDP $\rho$.
2: **Initialize Counter Arrays**
3: $\sigma \leftarrow \sqrt{\log(2/\beta)/\rho}$
4: $E \leftarrow \sqrt{\frac{2\log\frac{2}{\beta}}{\rho}} \cdot \sqrt{\log\frac{\frac{4}{\gamma}\log(\frac{2}{\beta})}{\beta}}$
5: **for** $r \leftarrow 1, 2, \ldots, \log(2/\beta)$ **do**
6:     **for** $c \leftarrow 1, 2, \ldots, 1/\gamma$ **do**
7:         $C[r, c] \leftarrow \mathcal{N}(0, \sigma^2)$ if Private CountSketch
8:         $C[r, c] \leftarrow E + \mathcal{N}(0, \sigma^2)$ if Private Count-Min
9:     **end for**
10: **end for**
11: **Output:** $C$.

---

In Algorithm 3, the set of arrays we use is $C$ which consists of $\log(2/\beta)$ arrays with length $1/\gamma$, which has the same space complexity as original Count-Min and CountSketch. Recall that two neighboring databases $X$ and $X'$ differ by at most one item. Therefore, after updating all the items respectively, for each corresponding array in $C(X)$ and $C(X')$, they differ by at most two elements and the difference is at most 1. Then the $\ell_2$-sensitivity of the set of arrays $C$ is bounded by

$$\Delta_2 = \sqrt{2\log(2/\beta)}. \tag{1}$$

By applying the Gaussian Mechanism (Definition 2.5), we can add *independent* Gaussian noises $\mathcal{N}(0, \sigma^2)$ to each counter in $C$, where $\sigma = \sqrt{\frac{\log(2/\beta)}{\rho}}$. Due to the privacy guarantee of Gaussian Mechanism (Lemma 2.8), it satisfies $\frac{\Delta_2^2}{2\sigma^2} = \rho$-zCDP.

Define $E(\beta, \gamma, \rho) = \sqrt{\frac{2 \log \frac{2}{\beta}}{\rho}} \cdot \sqrt{\log \frac{\frac{4}{\gamma} \log(\frac{2}{\beta})}{\beta}}$, for simplicity, we will use $E$ in Algorithm 3 and the proof in Appendix A. The private version of Count-Min can be derived by adding *independent* Gaussian noises $\mathcal{N}(E, \sigma^2)$ to each counter of $C$, while the private version of CountSketch can be derived by adding *independent* Gaussian noises $\mathcal{N}(0, \sigma^2)$ to each counter of $C$. The private versions of Count-Min and CountSketch are derived by combining Algorithm 3, Algorithm 1, and Algorithm 2.

## 3.1 Main results about Private Count-Min and CountSketch

We present the privacy guarantee and utility analysis of our Private Count-Min and CountSketch below. Recall that for each item $x$, we perform update as in Algorithm 1 and query as in Algorithm 2. In addition, $\widehat{f}(x)$ is the output estimated frequency and $f(x)$ is the actual frequency. To provide a bound for the additional error due to DP, we define $\widetilde{f}(x)$ to be the non-private estimated frequency (the output of the original Count-Min and CountSketch with the same set of hash functions). We begin with the properties of Private Count-Min. Note that all the proofs are deferred to Appendix A.

**Theorem 3.1.** *Private Count-Min satisfies $\rho$-zCDP regardless of the number of queries. Furthermore, with probability $1 - \beta$, the output $\widehat{f}(x)$ satisfies that*

$$\forall\, x, 0 \le \widehat{f}(x) - \widetilde{f}(x) \le 2E.$$

*In addition, for each item $x$, with probability $1 - \beta$,*

$$0 \le \widehat{f}(x) - f(x) \le \gamma \cdot N + 2E.$$

**Comparison to Count-Min.** Comparing our Theorem 3.1 with the conclusion in [Cormode and Muthukrishnan, 2005], our Private Count-Min preserves the nice property that the output will not underestimate the frequency with high probability. Furthermore, within the most popular regime where the privacy budget $\rho$ is a constant, the additional error bound due to differential privacy is independent of the size of database $N$, therefore it will become negligible as $N$ goes large.

**Justification of our Gaussian noise.** Note that with high probability, all the noises we add ($E + \sigma_{i,j}$, $\sigma_{i,j} \sim \mathcal{N}(0, \sigma^2)$) will be non-negative. Therefore, the noise we add and the original error induced by Count-Min will directly sum up and lead to larger error in evaluation. However, we claim that the additional $E$ ensures that with high probability, for all item $x$, the output will not underestimate the actual frequency. This nice property enables the good performance of our Private Count-Min when used in approximate top $k$ task, as shown in Section 5.

Next, Theorem 3.2 shows the properties of Private CountSketch.

**Theorem 3.2.** *Private CountSketch satisfies $\rho$-zCDP regardless of the number of queries. Furthermore, the frequency query from Private CountSketch is unbiased and with probability $1 - \beta$,*

$$\forall\, x, |\widehat{f}(x) - \widetilde{f}(x)| \le E.$$

*In addition, for each item $x$, with probability $1 - \beta$,*

$$|\widehat{f}(x) - f(x)| \le \gamma \cdot N + E.$$

**Comparison to CountSketch.** Comparing our Theorem 3.2 with the conclusion in [Charikar et al., 2002], our Private CountSketch preserves the nice property that the output will be an unbiased estimate of the frequency. This property enables our use of Private CountSketch in quantile estimation below (Section 4). Furthermore, within the most popular regime where the privacy budget $\rho$ is a constant, the additional error bound due to differential privacy is independent of the size of database $N$, thus it will become negligible as $N$ goes large.

## 3.2 The Uniform Bound of Additional Error

Theorem 3.1 and Theorem 3.2 show a uniform bound $\sup_x |\widehat{f}(x) - \widetilde{f}(x)| \le O(E)$ for linear sketches, which upper bounds the additional error imposed on the estimated frequency due to Differential Privacy guarantees. To derive the point-wise bound for $|\widehat{f}(x) - f(x)|$, we combine our result with the point-wise bound for $|\widetilde{f}(x) - f(x)|$ [Cormode and Muthukrishnan, 2005, Charikar et al., 2002] (note it is straightforward to apply other analyses on the point-wise bound [Minton and Price, 2014, Larsen et al., 2021] due to the triangle inequality). Moreover, in Appendix E, we demonstrate that our analysis provides the tight uniform bound when items are drawn from a large universe.

# 4 Private Quantile Sketches

In this section, we apply our Private CountSketch to state of the art quantile sketches in the turnstile model. Our private Dyadic CountSketch can estimate all the quantiles accurately at the same time while ensuring differential privacy.

## 4.1 Revisiting DCS

In [Wang et al., 2013], it is shown that DCS can return all $\gamma$-approximate quantiles with constant probability using space $O\left(\frac{1}{\gamma}\log^{1.5}U\log^{1.5}(\frac{\log U}{\gamma})\right)$. More specifically, the sketch structure here consists of $\log U$ CountSketches, each CountSketch uses a counter arrays $C$, which is $d \times w$ counters. The choice of $d, w$ follows $d = \Theta\left(\log(\frac{\log U}{\gamma})\right)$ and $w = O\left(\sqrt{\log U \log(\frac{\log U}{\gamma})}/\gamma\right)$.

## 4.2 Private DCS

In this work, we aim to estimate the quantiles accurately while preserving privacy. We do this by replacing CountSketch with PrivateCountSketch, which bases on the same structure as CountSketch discussed above. Given the privacy budget $\rho$, the privacy budget of each Private CountSketch is thus $\rho_0 = \frac{\rho}{\log U}$, due to composition of zCDP (Lemma 2.7). The $\ell_2$-sensitivity of each Private CountSketch is

$$\Delta_2 = O(\sqrt{2d}) = O\left(\sqrt{\log\left(\frac{\log U}{\gamma}\right)}\right).$$

To keep the whole algorithm $\rho$-zCDP (Lemma 2.7). Therefore, Gaussian Mechanism (Definition 2.5) with $\sigma^2 = O\left(\log U \log\left(\frac{\log U}{\gamma}\right)/\rho\right)$ ensures $\rho$-zCDP (Lemma 2.8). Similar to Lemma A.1, define $E(\gamma, U) = O\left(\sqrt{\frac{\log U \log(\frac{\log U}{\gamma})}{\rho}} \cdot \sqrt{\log \frac{\log U \log(\frac{\log U}{\gamma})}{\gamma}}\right)$, we can prove that with constant probability, all the Gaussian noises we add to all $\log U$ CountSketches are bounded by $E$ (for simplicity, we use $E$ to represent $E(\gamma, U)$).

Conditioned on the high probability event above, we prove that for a fixed quantile, the estimated quantile will be accurate with high probability. As has been proven in Theorem 3.2, the output estimated frequency is unbiased for any item. Therefore, similar to [Wang et al., 2013], for any item $x$ (corresponding to a fixed CountSketch), we have the output $\widehat{f}(x)$ of that CountSketch satisfies

$$\mathbb{P}\left[\left|\widehat{f}(x) - f(x)\right| > \frac{1}{w} \cdot N + E\right] < \exp\left(-O(d)\right) = O\left(\frac{\gamma}{\log U}\right).$$

By a union bound, with probability $1 - \log U \times O\left(\frac{\gamma}{\log U}\right) = 1 - O(\gamma)$, for any item corresponding to this fixed quantile, the error of CountSketch is bounded by $\frac{1}{w} \cdot N + E$. Conditioned upon this event, by Hoeffding's inequality, with probability $1 - O\left(\frac{\gamma}{\log U}\right)$, the sum of $\log U$ such independent errors is bounded by

$$\sqrt{\log U \log\left(\frac{\log U}{\gamma}\right)} \cdot \left(\frac{N}{w} + E\right) = \gamma \cdot N + E', \tag{2}$$

where $E' = O\left(\frac{\log U \log(\frac{\log U}{\gamma})}{\sqrt{\rho}} \cdot \sqrt{\log \frac{\log U \log(\frac{\log U}{\gamma})}{\gamma}}\right)$. To sum up, for a fixed quantile, with probability $1 - O(\gamma)$, the estimating error is bounded by $\gamma \cdot N + E'$.

Finally, apply another union bound on the $\frac{1}{\gamma}$ different quantiles, with constant probability, all the quantiles are estimated accurately (within the error bound (2)). Note that similar to [Wang et al., 2013], the failure probability here is a constant. For any failure probability $\beta$, we can further increase $d$ by a factor of $\log\frac{1}{\beta}$ to reduce this failure probability to $\beta$.

**Take-away of Private DCS.** First, our Private DCS has a same space complexity as the original DCS. In addition, according to (2), the additional error bound is proportional to $\frac{\log U \log \frac{1}{\gamma}}{\sqrt{\rho}}$ (ignoring $\log \log$ terms), and independent to the size of database $N$. In the most popular regime where the privacy budget $\rho$ is a constant, the additional error bound only appears as lower order terms, which will become negligible as $N$ goes large.

## 5 Evaluation

We have implemented DP linear sketches and DP DCS, and conducted extensive experiments to evaluated the privacy-utility trade-off of our proposed private sketches. The implementations are written in Python with the advantage of fast prototyping and good readability. The code for the following experiments can be found on Github [3].

### 5.1 Data Sets

The experimental evaluation is conducted using both synthetic and real world data sets. We consider the synthetic Zipf dataset Zipf [2016] with universe size of $2^{16}$ and the source IP addresses from CAIDA Anonymized Internet Trace 2015 dataset pas with universe size of $2^{32}$. For each independent run in the experiments, we use an input database size $N = 10^5$.

### 5.2 Metrics

In all experiments, we average the various metrics over 5 independent runs to minimize the measurement variance. The metrics used in the experiments are:

**Average Relative Error**: Let the set $\Psi$ denotes all unique items in the database. Average Relative Error (ARE) is computed based on $\Psi$ in which $\frac{1}{\Psi} \sum_{e \in \Psi} \frac{|f(e) - \hat{f}(e)|}{f(e)}$.

**F1 Score**: F1 score is the harmonic mean of the precision and recall ($2 \cdot \frac{precision \cdot recall}{precision + recall}$).

**Average Rank Error**: For each evenly spaced quantile and its associated item, we average the distance between the true rank and estimated rank.

We use ARE to evaluate sketch performance on frequency estimation and F1 score to evaluate the sketch's performance in identifying the top 10 items. For quantile approximation, we consider the $m$ evenly spaced quantiles and items. For instance, if $m = 1$, we consider the rank error for the median item; if $m = 2$, we consider then average rank error for the $33^{rd}$ and $67^{th}$ percentile items. Lower ARE and average rank error, and higher F1 score indicate better approximation.

### 5.3 Private Linear Sketches Experiments

To evaluate the utility of DP linear sketches, we compare the average relative error (ARE) and F1 score for frequency estimation and identify the top 10 items, respectively. As shown in Figure 1, the x-axis represents the space budget for each sketch (from 9.2 KB to 147.3 KB), and the y-axis denotes ARE or F1 score. The DP linear sketches use $\rho \in \{0.1, 1, 10\}$ in which lower $\rho$ value indicates more noise need to be added, and all sketches assume $\beta = 1\%$.

For frequency estimation, the performance of our private CountSketch with various privacy budgets is basically equivalent to the performance of the non-private CountSketch. Under different space and privacy budgets, they have minimal difference in ARE for both Zipf and CAIDA datasets, meaning that, while providing strong privacy guarantee, the estimated frequencies are still very accurate. The accurate estimation of private CountSketch is primarily due to the unbiased nature of CountSketch in which, by adding Gaussian noise, the private CountSketch still provides unbiased estimation for an item's frequency as proved in Theorem 3.2. As shown in both Figure 1(a) and Figure 1(b), the performance of private Count-Min degrades when the space budget increases or the privacy budget decreases. This behavior is expected as the upper bound on the frequency error in Theorem 3.1 has a dependency on both $\gamma$ and $\rho$. In order to preserve the property of not underestimating an item's

---

[3]https://github.com/ZhaoFuheng/Differentially-Private-Linear-Sketches

frequency, the private Count-Min sketch needs to add larger noise to each counter when the number of counters increases. As a result, the estimated frequencies for low-frequency items become inflated and this in turn decreases the overall accuracy.

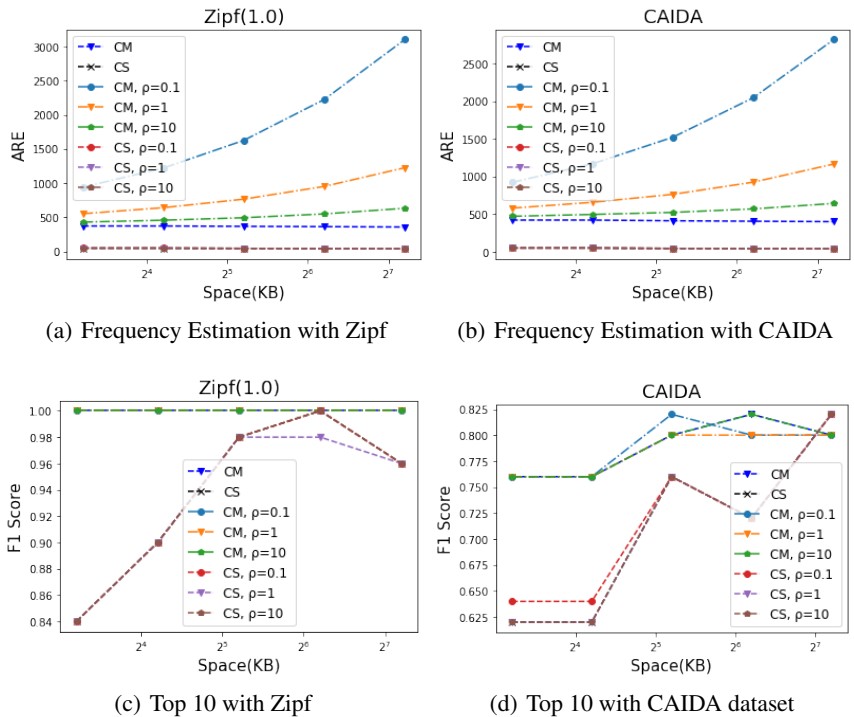

(a) Frequency Estimation with Zipf    (b) Frequency Estimation with CAIDA

(c) Top 10 with Zipf    (d) Top 10 with CAIDA dataset

Figure 1: Comparison of non-private linear sketches and DP linear sketches with various privacy budget under synthetic and real world datasets. The experiments assume $\beta = 1\%$ and $N = 10^5$.

For approximate top 10 items, private CountSketch has similar performance to CountSketch. Since both non-private and private CountSketch are unbiased, they may underestimate the frequency of true top K items and decrease the recall. On the other hand, the property of no underestimation is desirable for approximate top K items. In particular, non-private and private Count-Min sketch score high F1 scores for all datasets. While providing privacy guarantees, private Count-Min achieve 1.0 F1 scores for all space and all privacy budgets in Zipf dataset, as shown in Figure 1(c).

### 5.4 Private Quantile Sketch Experiments

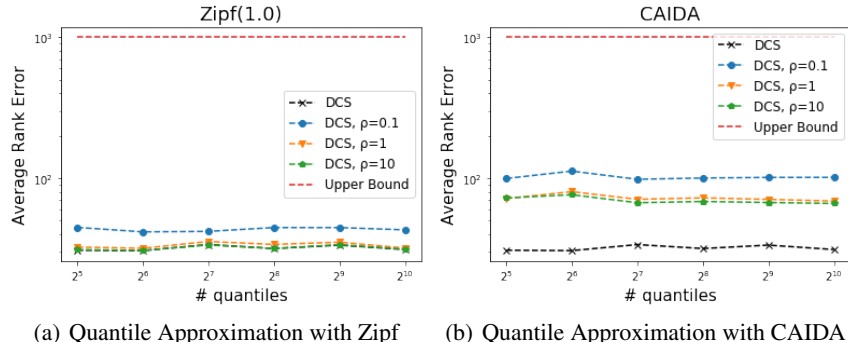

(a) Quantile Approximation with Zipf    (b) Quantile Approximation with CAIDA

Figure 2: Compare DCS and DP DCS with various privacy budget under synthetic and real world datasets. The experiments assume $\gamma = 1\%$, $N = 10^5$, and the desired error upper bound is $10^3$ ($\gamma N$).

To evaluate the utility of DP DCS, we compare the average rank error. As shown in Figure 2, the x-axis represents the number of evenly spaced quantiles, and the y-axis denotes the average rank error. The DP DCS use privacy budget $\rho \in \{0.1, 1, 10\}$ and all sketches assume $\gamma = 1\%$.

For the quantile approximation, we observe that the increase in the number of evenly spaced quantiles does not impact the average rank error, as shown in both Figure 2(a) and Figure 2(b). Since the CAIDA dataset universe size ($2^{32}$) is larger than Zipf dataset universe size ($2^{16}$), the average rank error in the CAIDA dataset is larger than the average rank error in the Zipf dataset. As shown in Equation (2), the error bound has a term depending on the universe size in which a large universe size leads to more error. When the privacy budget decreases, the average rank error increases as more noise needs to be added. Comparing DP DCS with strong privacy ($\rho = 0.1$) to DCS, the increase in rank error is relatively small compared to the database size of $10^5$. In addition, the desired rank error upper bound is $\gamma \cdot N = 10^3$ and all the rank errors are one order of magnitude lower.

# 6 Related Works

In this section, we discuss and compare our results to previous literature on Private Count-Min Sketch [Mir et al., 2011, Melis et al., 2015, Ghazi et al., 2019], and the concurrent work on Private CountSketch [Pagh and Thorup, 2022]. In fact, Pagh and Thorup [2022] and us both independently discovered the same algorithm for Private CountSketch with differences in the theoretical analysis. To the best of our knowledge, we are the first to present a DP quantile sketch in the turnstile model.

**Private Count-Min.** Mir et al. [2011] proposed to add Laplace noise into the Count-Min Sketch estimator to derive the number of heavy hitters with Pan-Privacy [Dwork et al., 2010]. Similarly, Melis et al. [2015] add independent Laplace noise to each counter of the sketch instead of the estimator. However, adding Laplace noise breaks the nice property of never underestimation in Count-Min. In contrast, our private Count-Min guarantees no underestimation with high probability. [Ghazi et al., 2019] added one-sided binomial noise into each counter of the sketch to preserve the property of no underestimation. However, using the Binomial mechanism inherently implies approximate differential privacy [Canonne et al., 2020]. In contrast, by using the Gaussian mechanism, our Private Count-Min provides the stronger concentrated differential privacy guarantee.

**Private CountSketch.** Pagh and Thorup [2022] and us both independently discovered the same algorithm for private CountSketch. There is a major difference in the analysis and we believe both analyses are valuable, in which Pagh and Thorup [2022] focused on deriving a tight point-wise bound for $|\widehat{f}(x) - f(x)|$, while we focused on deriving a uniform bound for $\sup_x |\widehat{f}(x) - \widetilde{f}(x)|$. Our uniform bound for $\sup_x |\widehat{f}(x) - \widetilde{f}(x)|$ can derive the point-wise bound for $|\widehat{f}(x) - f(x)|$, by combining our result with any point-wise bound for $|\widetilde{f}(x) - f(x)|$ due to the triangle inequality. [Pagh and Thorup, 2022] obtains a tighter point-wise bound by using concentration of median instead of triangle inequality. However, Pagh and Thorup [2022]'s analysis can not imply the point-wise bound for $|\widehat{f}(x) - \widetilde{f}(x)|$. More detailed comparisons are included in Appendix C.

# 7 Conclusion

In this work, we demonstrate that linear sketches can be made differentially private and provide useful information while maintaining their original properties by adding a small amount of Gaussian noise at initialization. In addition, leveraging the private CountSketch, we propose the DP DCS for quantile approximation in the turnstile model. DP DCS achieves low rank errors even for a large data universe. Moreover, for all the proposed algorithms, when the privacy budget is constant, the additional error due to privacy is independent of the database size and the error will become negligible when the database grows larger. Moreover, private linear sketches bring new opportunities for other statistical questions such as the private euclidean distance estimation [Stausholm, 2021] which can be calculate as the dot product of two private linear sketches. As a result, we believe our proposed algorithms are efficient and practical for real-world systems and enable these systems to perform data analysis and machine learning tasks privately.

## Acknowledgments and Disclosure of Funding

This work is partially supported by gifts from Snowflake Inc, and NSF grants CNS-1703560, CNS-1815733 and CNS-2048091. The authors thank Rasmus Pagh for a helpful discussion regarding their concurrent work [Pagh and Thorup, 2022]. The authors also thank Adam Smith for clarifying the mergeability in the inherently private Flajolet-Martin Sketch [Smith et al., 2020].

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
