# A  Missing proofs

In this section, we present the missing proofs. Recall that for each item $x$, we perform query as in Algorithm 2. In the proof below, we use $\mathrm{arr}'$ to denote the $\mathrm{arr}$ in Algorithm 2 (with private initialization) while $\mathrm{arr}$ denotes the $\mathrm{arr}$ in Algorithm 2 under non-private initialization (all zero initialization) and the same set of hash functions. In addition, $\widehat{f}(x)$ is the output estimated frequency, $f(x)$ is the actual frequency and $\widetilde{f}(x)$ is the non-private estimated frequency (the output of the original Count-Min and CountSketch with the same set of hash functions). We first present the following Lemma A.1, which gives a high probability bound for the Gaussian noises we add.

**Lemma A.1** (Utility analysis). *If there are $\frac{1}{\gamma}\log(\frac{2}{\beta})$ independent Gaussian noises sampled from $\mathcal{N}(0, \sigma^2)$ (where $\sigma = \sqrt{\frac{\log(2/\beta)}{\rho}}$), denoted as $\sigma_{ij}$ where $i \in [\log(2/\beta)]$, $j \in [1/\gamma]$, then with probability $1 - \frac{\beta}{2}$, for any $i \in [\log(2/\beta)]$, $j \in [1/\gamma]$,*

$$|\sigma_{ij}| \le \sqrt{2}\sigma \cdot \sqrt{\log \frac{\frac{4}{\gamma}\log(\frac{2}{\beta})}{\beta}} = \sqrt{\frac{2\log\frac{2}{\beta}}{\rho}} \cdot \sqrt{\log \frac{\frac{4}{\gamma}\log(\frac{2}{\beta})}{\beta}}. \tag{3}$$

*Proof of Lemma A.1.* The lemma directly results from the concentration inequality of Gaussian distribution and a union bound. □

**Theorem A.2** (Restate Theorem 3.1). *Private Count-Min satisfies $\rho$-zCDP regardless of the number of queries. Furthermore, with probability $1 - \beta$, the output $\widehat{f}(x)$ satisfies that*

$$\forall x, 0 \le \widehat{f}(x) - \widetilde{f}(x) \le 2E.$$

*In addition, for each item $x$, with probability $1 - \beta$,*

$$0 \le \widehat{f}(x) - f(x) \le \gamma \cdot N + 2E.$$

*Proof of Theorem A.2.* Differential privacy directly results from the DP guarantee of Gaussian Mechanism (Lemma 2.8) and post processing (Lemma 2.7).

According to Lemma A.1, we have with probability $1 - \frac{\beta}{2}$, for all noises $E + \sigma_{ij}$, where $\sigma_{ij} \sim \mathcal{N}(0, \sigma^2)$, $i \in [\log(2/\beta)]$, $j \in [1/\gamma]$, it holds that

$$0 \le E + \sigma_{ij} \le 2E.$$

Under the above case that will happen with probability $1 - \frac{\beta}{2}$, for all $x$ and the corresponding $\mathrm{arr}$, $\mathrm{arr}'$, it holds that

$$\min(\mathrm{arr}) \le \min(\mathrm{arr}') \le \min(\mathrm{arr} + 2E \cdot \mathbb{1}) \le \min(\mathrm{arr}) + 2E.$$

Therefore, for all $x$,

$$\widetilde{f}(x) \le \widehat{f}(x) \le \widetilde{f}(x) + 2E.$$

For the last conclusion (point-wise bound), by the property of Count-Min [Cormode and Muthukrishnan, 2005], we have for any item $x$, with probability $1 - \frac{\beta}{2}$,

$$0 \le \min(\mathrm{arr}) - f(x) \le \gamma \cdot N.$$

Therefore, for any item $x$, with probability $1 - \beta$, we have

$$\widehat{f}(x) = \min(\mathrm{arr}') \ge \min(\mathrm{arr}) \ge f(x)$$

and

$$\widehat{f}(x) = \min(\mathrm{arr}') \le \min(\mathrm{arr}) + 2E \le f(x) + \gamma \cdot N + 2E.$$

Then the proof is completed by plugging in the definition of $E$. □

**Theorem A.3** (Restate Theorem 3.2). *Private CountSketch satisfies $\rho$-zCDP regardless of the number of queries. Furthermore, the frequency query from Private CountSketch is unbiased and with probability $1 - \beta$,*

$$\forall\, x, |\widehat{f}(x) - \widetilde{f}(x)| \leq E.$$

*In addition, for each item $x$, with probability $1 - \beta$,*

$$|\widehat{f}(x) - f(x)| \leq \gamma \cdot N + E.$$

*Proof of Theorem A.3.* First of all, differential privacy directly results from the DP guarantee of Gaussian Mechanism (Lemma 2.8) and post processing (Lemma 2.7).

Next we claim that the conclusion that $\mathbb{E}\widehat{f}(x) = f(x)$ arises from symmetry. If we replace $\{h_i\}_{i \in [\log(2/\beta)]}$, $\{g_i\}_{i \in [\log(2/\beta)]}$, $\{\sigma_{ij}\}_{i \in [\log(2/\beta)], j \in [1/\gamma]}$ with $\{h_i\}_{i \in [\log(2/\beta)]}$, $\{g_i'\}_{i \in [\log(2/\beta)]}$, $\{-\sigma_{ij}\}_{i \in [\log(2/\beta)], j \in [1/\gamma]}$ where $g_i'(x) = g_i(x)$ and $g_i'(x') = -g_i(x')$, $\forall\, x' \neq x$, then the outputs under these two cases will be symmetric around $f(x)$ and the probability distribution function at these two cases are identical. Therefore, we have

$$\mathbb{E}\widehat{f}(x) = f(x). \tag{4}$$

According to Lemma A.1, we have with probability $1 - \frac{\beta}{2}$, for all noises $\sigma_{ij} \sim \mathcal{N}(0, \sigma^2)$, $i \in [\log(2/\beta)]$, $j \in [1/\gamma]$, it holds that

$$|\sigma_{ij}| \leq E.$$

Under the above case that will happen with probability $1 - \frac{\beta}{2}$, we will prove that for all $x$,

$$|\widehat{f}(x) - \widetilde{f}(x)| \leq E.$$

Without loss of generality, we can assume that $\log \frac{2}{\beta} = 2k + 1$, where $k$ is a positive integer (we can choose $k$ to be the minimum integer such that $2k + 1 \geq \log \frac{2}{\beta}$).

Suppose that $\mathrm{median}(\mathrm{arr}') > \mathrm{mediam}(\mathrm{arr}) + E$, then it holds that there are at least $k + 1$ elements in $\mathrm{arr}'$ that is larger than $\mathrm{mediam}(\mathrm{arr}) + E$ due to the definition of median. Because $|\sigma_{ij}| \leq E$, for all $i, j$, there are at least $k + 1$ elements in $\mathrm{arr}$ that is larger than $\mathrm{mediam}(\mathrm{arr})$, which leads to contradiction. As a result, we have

$$\widehat{f}(x) - \widetilde{f}(x) = \mathrm{median}(\mathrm{arr}') - \mathrm{mediam}(\mathrm{arr}) \leq E.$$

Similarly, $\widehat{f}(x) - \widetilde{f}(x) \geq -E$. Combining these two results, for all $x$, it holds that

$$|\widehat{f}(x) - \widetilde{f}(x)| \leq E.$$

Finally, we prove the point-wise bound. By the property of CountSketch [Charikar et al., 2002], we have for any item $x$, with probability $1 - \frac{\beta}{2}$,

$$|\widetilde{f}(x) - f(x)| \leq \gamma \cdot N.$$

Therefore, according to triangle inequality, for each item $x$, with probability $1 - \beta$,

$$|\widehat{f}(x) - f(x)| \leq \gamma \cdot N + E.$$

Then the proof is completed by plugging in the definition of $E$. $\qquad\square$

## B   Missing Quantile Algorithms

Gilbert et al. [2002] made the connection between frequency and quantiles to propose the first universe based $RSS$ quantile sketch in the turnstile model. Observe, that the relationship between frequency and rank is that one can sum up all items' frequency in the range of 0 to the item itself to estimate the rank. However, this naive approach requires summing all items' frequencies in the range, and the error quickly escalates. A better approach is to break the range from 0 to item $x$ into at most $\log U$ dyadic intervals [Cormode et al., 2019] and then sum all frequencies for each dyadic interval to obtain the estimation of the rank$(x)$. Cormode and Muthukrishnan [2005] proposed the Dyadic Count-Min

sketch (DCM) which uses Count-Min sketches for estimating the frequencies of each dyadic interval with space complexity $O(\frac{1}{\gamma} \log^2 U \log \frac{\log U}{\gamma})$ and update time $O(\log U \log \frac{\log U}{\gamma})$. Later, Wang et al. [2013] leveraged the unbiased property of CountSketch and proposed the Dyadic CountSketch (DCS) which replaces the Count-Min sketch with the CountSketch [Charikar et al., 2002] to further improve the space complexity to $O(\frac{1}{\gamma} \log^{1.5} U \log^{1.5}(\frac{\log U}{\gamma}))$ while maintaining the same update time. DCS and DCM share the same update and query algorithms as shown below. DCS uses $O(\frac{1}{\gamma} \log U \log \frac{\log U}{\gamma})$ space for each Count-Min sketch and DCS uses $O(\frac{1}{\gamma} \log^{0.5} U \log^{1.5}(\frac{\log U}{\gamma}))$ space for each CountSketch, where both of them use $O(\log \frac{\log U}{\gamma})$ rows. For more specific details, [Cormode and Yi, 2020] provide a comprehensive analysis of linear and quantile sketches.

---

**Algorithm 4** DCS/DCM Update$(x, v)$

---

1: **Input:** Item $x$ with value $v \in \{-1, +1\}$, and an array of linear sketches $\{LS_0, \ldots, LS_{\log U}\}$.
2: **for** $j \leftarrow 0, \ldots, \log U$ **do**
3:     $LS_j$.update$(x, v)$
4:     $x \leftarrow \lfloor x/2 \rfloor$
5: **end for**
6: **Output:** $\{LS_0, \ldots, LS_{\log U}\}$.

---

---

**Algorithm 5** DCS/DCM Query$(x)$

---

1: **Input:** Item $x$, and an array of linear sketches $\{LS_0, \ldots, LS_{\log U}\}$.
2: $R \leftarrow 0$
3: **for** $i \leftarrow 0, \ldots, \log U$ **do**
4:     **if** $x$ is odd **then**
5:         $R \leftarrow R + LS_j$.query$(x)$
6:     **end if**
7:     $x \leftarrow \lfloor x/2 \rfloor$
8: **end for**
9: **Output:** $R$.

---

Based on the observations, DCS and DCM quantile sketches keeps $\log U$ number of linear sketches, one for each dyadic interval. As a result, to update an item $x$ with value $v \in \{-1, +1\}$, DCS and DCM need to update $\log U$ levels: they first map item $x$ to a dyadic interval for the level and then update the corresponding linear sketch, as shown in Algorithm 4. To estimate the rank of an item, DCS and DCM first break the range into at most $\log U$ dyadic intervals and then query the frequency for each interval from the corresponding linear sketch, as shown in Algorithm 5.

## C   Detailed Comparison for Private CountSketch

Recall that for some item $x$, $\widehat{f}(x)$ is the output estimated frequency, $f(x)$ is the actual frequency and $\widetilde{f}(x)$ is the non-private estimated frequency (the output of the original Count-Min and CountSketch with the same set of hash functions). Let $k$ denote the number of rows in our counter, and we choose $k = \log \frac{2}{\beta}$ to bound the failure probability by $\beta$. We first state both Pagh and Thorup [2022]'s results and ours.

Our uniform bound (Theorem 3.2): $\sup_x |\widehat{f}(x) - \widetilde{f}(x)| \leq E = \widetilde{O}(\sqrt{\frac{k}{\rho}})$.

Our lower bound (Theorem E.1): $\sup_x |\widehat{f}(x) - \widetilde{f}(x)| \geq \Omega(\sqrt{\frac{k}{\rho}})$.

Pagh and Thorup [2022]'s point-wise bound[4]: $|\widehat{f}(x) - f(x)| \leq$ non-private error bound $+ \widetilde{O}(\sqrt{\frac{1}{\rho}})$.

There is a major difference between the analysis in [Pagh and Thorup, 2022] and ours. While Pagh and Thorup [2022] focused on the point-wise bound for $|\widehat{f}(x) - f(x)|$, we focused on the uniform bound for $\sup_x |\widehat{f}(x) - \widetilde{f}(x)|$. We are interested in the trade-off between privacy and accuracy for the

---

[4]We reformulate the bound in [Pagh and Thorup, 2022] for comparison.

CountSketch. In particular, we want to answer the question of what additional error will be imposed on the estimated frequency due to the Differential Privacy guarantee. As a result, our result shows an uniform bound $\sup_x |\widehat{f}(x) - \widetilde{f}(x)| \leq E$ where $E$ is a function of the desired accuracy, failure probability, and privacy guarantee. To derive the point-wise bound for $|\widehat{f}(x) - f(x)|$, we can simply combine our result with any point-wise bound for $|\widetilde{f}(x) - f(x)|$ (like the one in our work or [Pagh and Thorup, 2022]) due to triangular inequality. However, Pagh and Thorup [2022]'s analysis can not imply even point-wise bound for $|\widehat{f}(x) - \widetilde{f}(x)|$.

We agree that Pagh and Thorup [2022] has a tight point-wise bound for $|\widehat{f}(x) - f(x)|$ by using the concentration of median. By comparing our analysis and [Pagh and Thorup, 2022] for $|\widehat{f}(x) - f(x)|$, Pagh and Thorup [2022]'s point-wise bound removes the $\sqrt{\log \frac{2}{\beta}}$ ($\beta$ denotes failure probability) in the lower order term $E$ which is added to the original estimation error $|\widetilde{f}(x) - f(x)|$. However, the difference may not be substantial. When the database is large (which is the usual case for why we need to perform approximations), $E$ is small compared to the $|\widetilde{f}(x) - f(x)|$. Even for the extreme case of setting $\beta = 1e - 10$, the amplification factor for calculating $E$ is $\sqrt{\log \frac{2}{\beta}} \approx 5.8$, which $E$ is still very likely to be small compare to $|\widetilde{f}(x) - f(x)|$.

We believe that none of the two results dominate each other. Both Pagh and Thorup [2022]'s point-wise bound for $|\widehat{f}(x) - f(x)|$ and our uniform bound for $|\widehat{f}(x) - \widetilde{f}(x)|$ are useful analysis for understanding the Differentially Private CountSketch with Gaussian noise.

# D    Extension to the Data Stream Setting

Our Differentially Private Linear Sketches only guarantee Differential Privacy for the query of database, i.e., the adversary is only allowed to query after the whole database passes our algorithm. However, queries for data stream is also quite practical in real-life applications, i.e., the adversary can query at any time. Take reinforcement learning (RL) as an instance, we can use differentially private linear sketches to estimate the visitation number of all (state,action) pairs while preserving privacy. If we only have linear sketches for database, we can only handle offline RL [Qiao and Wang, 2022b], while with linear sketches for data stream, we can deal with the more challenging online RL [Qiao and Wang, 2022a, Qiao et al., 2022].

Our algorithms can be extended to the data stream setting with moderate modifications. Different from our approach of adding noise at the beginning (Algorithm 3), we need to add noise after each item passes our algorithm. To guarantee Differential Privacy under data stream, we can apply the tree-based algorithm (as shown in Kairouz et al. [2021]) to add Gaussian noises to continuous data. In this way, the algorithm is Differentially Private no matter how many times the adversary queries the data stream and the additional error bound is the same scale as $E$ in our main theorems, with some extra multiplicative logarithmic terms.

# E    Lower Bound for the Additional Error due to Privacy

In this section, we provide a lower bound for CountSketch (the counterpart for Count-Min is similar and we omit it here) showing that our analysis of $\sup_x |\widehat{f}(x) - \widetilde{f}(x)|$ is tight. For simplicity, we assume the counter arrays $C$ we use has shape $k \times d$ and $k$ is odd, which means our upper bound $E$ for $\sup_x |\widehat{f}(x) - \widetilde{f}(x)|$ is $\widetilde{O}(\sqrt{\frac{k}{\rho}})$. Then according to our Algorithm 3, the noise we add has scale (standard variance) $\sqrt{\frac{k}{\rho}}$. The following theorem shows that if the universe is large enough, with constant probability, $\sup_x |\widehat{f}(x) - \widetilde{f}(x)| \geq \Omega(\sqrt{\frac{k}{\rho}})$, which matches our upper bound in Theorem 3.2 up to logarithmic terms.

**Theorem E.1.** *There exists constants $c, p$, such that if the size of universe satisfies $U \geq ckd(1 + \frac{1}{d-1})^{k-1}$, for our Private CountSketch, there exists a database, with probability at least $p$,*

$$\sup_x |\widehat{f}(x) - \widetilde{f}(x)| \geq \sqrt{\frac{k}{\rho}}.$$

*Proof of Theorem E.1.* Fix some item $x$ with its corresponding $\{h_i(x), g_i(x)\}_{i \in [k]}$. For each $i \in [k]$, we aim to find $y_i$ in universe such that $h_i(y_i) = h_i(x)$, $h_j(y_i) \neq h_j(x)$, $\forall j \neq i$ and $g_i(y_i) = g_i(x)$ (this is replaced with $g_i(y_i) = -g_i(x)$ if $i \geq \frac{k+1}{2}$). For any item in the universe, due to the uniform randomness of its hash functions, the probability it satisfies such conditions is $Pr = \frac{1}{2d}(1 - \frac{1}{d})^{k-1}$. Therefore, when $U \geq ckd(1 + \frac{1}{d-1})^{k-1}$ for some constant $c$, with constant probability, we can find $\{y_i\}_{i \in [k]}$ satisfying the previous conditions.

Next we can construct a database with only $\{x, y_1, \cdots, y_{k-1}\}$, where the frequency of $x$ is some large $n_x$, and the frequencies of all $y_i$'s are $n_y$ that satisfies that $n_y$ is much larger than $E$. Therefore, the arr of $x$ (without adding noise) consists of one $n_x$, $\frac{k-1}{2}$ numbers much larger than $n_x + E$ and $\frac{k-1}{2}$ numbers much smaller than $n_x - E$. Finally, with high probability, the change from $\widetilde{f}(x)$ to $\widehat{f}(x)$ is from $n_x$ to $n_x + \mathcal{N}(0, \frac{k}{\rho})$, which finishes our proof. $\qquad \square$