# OpenReview forum: "Differentially Private Linear Sketches: Efficient Implementations and Applications"
_NeurIPS.cc/2022/Conference — NeurIPS 2022 Accept_

### Official Review · Reviewer_FaJp · 2022-06-25

**Rating:** 4
**Confidence:** 3
**Soundness:** 3 good
**Presentation:** 2 fair
**Contribution:** 2 fair

**Summary:**

The paper considers differential privacy for some standard sketches, determining their differential privacy bounds under Gaussian noise.

**Questions:**

I believe there are other similar (contemporaneous?) works on the arxiv.  See https://arxiv.org/abs/2205.08397.
Perhaps you can compare with this paper.

**Strengths And Weaknesses:**

Strength:

Natural problem from differential privacy.
Analysis + experiments.
Good set of experimental results.

Weaknesses:
Analysis is straightforward.
Doesn't get same one-sided error bounds just a "with high probability" variant.

Originality:  I'm concerned about the originality.  In some sense, similar results (with perhaps weaker or incomparable results?, it's hard to tell) appear in other papers, such as
Mir, D., Muthukrishnan, S., Nikolov, A., and Wright, R. N. Pan-private algorithms via statistics on
sketches. In Proceedings of Symposium on Principles of database systems (PODS), pp. 37–48,
2011. (which is cited in this paper)
and
Melis, L., Danezis, G., and Cristofaro, E. D. Efficient private statistics with succinct sketches. In
Annual Network and Distributed System Security Symposium (NDSS). The Internet Society, 2016. (which is not).
The results seem to follow just by applying the standard Gaussian mechanism and then applying composition theorems from other work.  (Lemmas 2.7-2.9).  Perhaps nobody has written this down before, but I have trouble believing it was not known as it seems a straighforward composition.

Quality:  Given the originality concerns above, the quality is OK.

Clarity:  The writing is generally good and clear.

Significance:  See the originality issues above.

---

> ### Author Response · Authors · 2022-07-31
> **Response to Reviewer FaJp**
>
> Thank you very much for your time and constructive criticism! We absolutely agree that we should include discussions comparing our work and other related works, and especially the concurrent work [1] as mentioned in the feedback. In the revision, we present an improved analysis of our algorithm, showcase that our algorithm provide the tight uniform bound (Appendix E), and added detailed comparisons between our work and others in the new related work section. In addition, we also included a discussion on applying our algorithm in the stream setting (Appendix D).
>
> For private Count-Min, [2] proposed a new Pan-private estimator to estimate the number of heavy hitters using Count-Min by adding Laplace noise to the estimator. Similarly [3] added Laplace noise to each counter in Count-Min which breaks the nice property of never underestimation in Count-Min. In contrast, our private Count-Min guarantee of no underestimation with high probability. [4] added one-sided binomial noise into each counter of the sketch to preserve the property of no underestimation. However, using the Binomial mechanism inherently implies an approximate differential privacy [5]. In contrast, by using Gaussian mechanism, our Private Count-Min provides the stronger concentrated differential privacy guarantee.
>
> For private Count-Median, while [1] and our work proposed the same algorithm for private Count-Median, there are major differences in analysis between [1] and our work. We believe that both analyses are valuable for understanding the algorithm. In summary, [1] focused on the point-wise bound for $|\widehat{f}(x)-f(x)|$ , we focused on the uniform bound for $\sup_{x}|\widehat{f}(x)-\widetilde{f}(x)|$, where $f(x)$ denotes the true frequency of item $x$, $\widetilde{f}(x)$ denotes the non-private estimate of item $x$, and $\widehat{f}(x)$ denotes the private estimate of item $x$ (private and non-private Count-Median with the same set of hash functions). Our uniform bound for $\sup_{x}|\widehat{f}(x)-\widetilde{f}(x)|$ can derive the point-wise bound for $|\widehat{f}(x)-f(x)|$, by combining our result with any point-wise bound for $|\widetilde{f}(x)-f(x)|$ due to the triangle inequality. In addition, the uniform bound of our algorithm is tight when universe size is large which is often the case in the Big Data era. However, [1]'s analysis can not even imply the point-wise bound for $|\widehat{f}(x)-\widetilde{f}(x)|$. We agree that [1] provides a tight point-wise bound for $|\widehat{f}(x)-f(x)|$ by saving the low order term for the added estimation error due to privacy guarantees. For a large database, the added estimation error due to privacy is often small and the savings in analysis may not make a substantial difference. For linear sketch with $k$ rows:
>
> Our uniform bound (Theorem 3.2): $\sup_x |\widehat{f}(x)-\widetilde{f}(x)|\leq E=\widetilde{O}(\sqrt{\frac{k}{\rho}})$.
>
> Our lower bound (Theorem E.1): $\sup_x |\widehat{f}(x)-\widetilde{f}(x)|\geq \Omega(\sqrt{\frac{k}{\rho}})$.
>
> The point-wise bound in [1] (reformulated for comparison.): $|\widehat{f}(x)-f(x)|\leq \text{non-private error bound} + \widetilde{O}(\sqrt{\frac{1}{\rho}})$.
>
>
> [1] Improved Utility Analysis of Private Count Sketch, Pagh et al.
>
> [2] Pan-private algorithms via statistics on sketches. Mir et al.
>
> [3] Efficient private statistics with succinct sketches. Melis et al.
>
> [4] On the power of multiple anonymous messages. Ghazi et al.
>
> [5] The discrete gaussian for differential privacy. Canonne et al.

---

### Official Review · Reviewer_fzjL · 2022-07-11

**Rating:** 6
**Confidence:** 4
**Soundness:** 4 excellent
**Presentation:** 4 excellent
**Contribution:** 2 fair

**Summary:**

The paper present differentially private variants of linear sketches such as Count Min and Count Median sketches. The differential privacy guarantees are provided using a recent notion of differential privacy: zDP  [Dwork and Rothblum, 2016, Bun and Steinke, 2016]. The only change from vanilla sketches is the initialization provided in algorithm 3, which when combined with the update (Algorithm 1) and query (Algorithm 2) components of Count Min and Count median sketches leads to the differentially private versions of these sketches. The initialization itself adds Gaussian noise to each array item. Any other noise source satisfying zCDP would also suffice. The analysis of the algorithms directly follows by combining the analysis for vanilla sketches with the recent differentially privacy results.

 Building on the Differentially Private Count Median Sketch, the authors propose a Dyadic Count-Median sketch that can estimate all the quantiles simultaneously while ensuring differential privacy. The experimental evaluations validate the practicality of the algorithms.

Overall, while the algorithms presented, as well as their analysis, are not novel. The paper does a good job of reviewing the literature in two disjoint fields and providing a comprehensive set of differential private linear sketches, which need to be documented. Therefore, I recommend acceptance.

**Questions:**

Can the authors comment on the lower bounds for the various problems considered? If not, can the authors comment on the challenges in coming up with such a result?

**Limitations:**

Yes.

**Strengths And Weaknesses:**


1) The authors systematically study the literature on linear sketches and differential privacy and provide a comprehensive set of differentially private linear sketches
 2) The paper is well written and easy to follow.
3) However, there isn't any novelty either algorithmically or in the theoretical analysis of the algorithms: The only real change from an algorithm standpoint is the initialization phase of the linear sketches. The proofs follow by bootstrapping the standard proofs for linear sketches and using the recent differentially privacy results.
4) This would be a solid paper had the authors presented some sort of impossibility result for frequency estimation under privacy guarantees.

---

> ### Author Response · Authors · 2022-07-31
> **Response to Reviewer fzjL**
>
> We really appreciate the positive feedback given and agree that deeper analysis should be performed on the algorithm. In the revision, we present an improved analysis of our algorithm, showcase that our algorithm provide the tight uniform bound (Appendix E), and add a new section to compare our algorithm with related works. In the response to Reviewer a3xv, we explained the major difference in analysis between [1] and our work. We believe that both analysis are valuable for understanding the algorithm. In summary, [1] focused on the point-wise bound for $|\widehat{f}(x)-f(x)|$ , we focused on the uniform bound for $\sup_{x}|\widehat{f}(x)-\widetilde{f}(x)|$, where $f(x)$ denotes the true frequency of item $x$, $\widetilde{f}(x)$ denotes the non-private estimate of item $x$, and $\widehat{f}(x)$ denotes the private estimate of item $x$ (private and non-private Count-Median with the same set of hash functions). Our uniform bound for $\sup_{x}|\widehat{f}(x)-\widetilde{f}(x)|$ can derive the point-wise bound for $|\widehat{f}(x)-f(x)|$, by combining our result with any point-wise bound for $|\widetilde{f}(x)-f(x)|$ due to the triangle inequality. However, [1]'s analysis can not even imply the point-wise bound for $|\widehat{f}(x)-\widetilde{f}(x)|$. We agree that [1] provides a tight point-wise bound for $|\widehat{f}(x)-f(x)|$ by saving a lower order term for the added estimation error due to privacy guarantees. For a large database, the added estimation error due to privacy is often small and the saving in analysis may not make a substantial difference. For linear sketch with $k$ rows:
>
> Our uniform bound (Theorem 3.2): $\sup_x |\widehat{f}(x)-\widetilde{f}(x)|\leq E=\widetilde{O}(\sqrt{\frac{k}{\rho}})$.
>
> Our lower bound (Theorem E.1): $\sup_x |\widehat{f}(x)-\widetilde{f}(x)|\geq \Omega(\sqrt{\frac{k}{\rho}})$.
>
> The point-wise bound in [1] (reformulated for comparison.): $|\widehat{f}(x)-f(x)|\leq \text{non-private error bound} + \widetilde{O}(\sqrt{\frac{1}{\rho}})$.
>
> Moreover, to strengthen our contributions, in the revision, we added an appendix discussing Continuous Release in a stream setting. Previously, DP linear sketch algorithms only guarantees Differential Privacy when querying a database, i.e., the adversary is only allowed to query after the whole database passes our algorithm. However, queries for data stream are essential for real-life applications, i.e., the adversary can query at any time. Our algorithm can be extended to this setting with moderate modifications. Unlike our original approach of adding noise at the beginning, we need to add noise at each step. To guarantee Differential Privacy under a data stream setting, we can apply the tree-based algorithm (as shown in [2]) to add Gaussian noises to continuous data. In this way, the algorithm is DP no matter how many times the adversary queries the data stream and the additional error bound is the same scale as $E$ in our work, with some additional multiplicative logarithmic terms. The algorithm and analysis for data stream is added in the appendix of the revision.
>
> [1] Improved Utility Analysis of Private Count Sketch, Pagh et al.
>
> [2] Practical and Private (Deep) Learning Without Sampling or Shuffling. Peter Kairouz et al.

---

### Official Review · Reviewer_kKXq · 2022-07-12

**Rating:** 7
**Confidence:** 2
**Soundness:** 3 good
**Presentation:** 3 good
**Contribution:** 3 good

**Summary:**

This paper demonstrates that linear sketches can be made differentially private by adding a small amount of Gaussian noise at initialization. Experimental results on Zipf and the real CAIDA dataset show that the utility is less influenced by this modification.

**Questions:**

W1 and W3.

**Limitations:**

The authors adequately addressed the limitations and potential negative societal impact of their work.

**Strengths And Weaknesses:**

Strengths:

- Trendy topic
- Well-organized paper
- The improvement is simple yet effective

Weaknesses:

- The assumption limits the application scenarios
- Lack of analysis

Detailed Weakness Comments:

This paper points out that linear sketches can be made differentially private and provide useful information while maintaining their original properties. Specifically, contrary to the standard operation that initializes each counter with 0, the authors initialize each counter with a noise sampled from certain Gaussian distribution, whose parameters are determined by the accuracy parameter, failure probability, and budget for zCDP.

I like this idea overall, though this modification is small, the authors prove that this could make the linear sketches differentially private. However, this paper also has some flaws, such as the assumption, and lacking analysis of the case where the adversary is actively involved. I list my concrete concerns in the following:

W1. The assumption limits the application scenarios. In section 2.2, the authors briefly introduce linear sketches, they assume that item x has value v equals either -1 or 1, which limits the application scenarios of linear sketches. However, from algorithms 1 and 2, I didn’t see any operation that relies on this assumption. Therefore, I would like to know whether this is a redundant assumption, or if this assumption is required to make sure the private linear sketches satisfy differential privacy. If this assumption is only needed to achieve the DP guarantee, I think the trade-off between privacy and applicability should be discussed.

W2. It would be nice if the authors could provide a brief illustration about why vanilla linear sketch does not hold differential privacy property, which may make the improvement intuition clearer in this paper.

W3. Lack of analysis. The key idea to making existing linear sketches private is to initialize counters with random noise, as shown In algorithm 3. And the authors provide a series of analysis to prove that the modified scheme satisfies the DP definition, which is nice. However, I wonder whether these properties still hold when the private linear sketches are attacked. For example, the authors initialize counter C with E plus noise sampled from a gaussian distribution (for count-min case), yet, the attacker could eliminate this E in the initialization step, since E is determined once the accuracy parameter, failure probability, and budget for zCDP are fixed. I would like to know would the scheme still satisfy the DP requirement?

---

> ### Author Response · Authors · 2022-07-31
> **Response to Reviewer kKXq**
>
> Thanks a lot for the positive review and valuable feedback! In the response, we address your points individually. Moreover, in our revised version of the paper, we present an improved analysis of our algorithm, showcase that our algorithm provide the tight uniform bound (Appendix E), include a new section to discuss and compare our algorithms with other related works, and discuss our algorithm in the stream setting (Appendix D).
>
> W1: In this paper, we assume the unit update model which is prevalent in real world applications such as analysis of click stream and database inserts and deletes based on primary keys. The value $v$ corresponds to the addition or removal action of a particular item.
>
> W2: Thanks for the great suggestion. Due to space constraints for the paper, there isn't enough space to include reasons for why vanilla linear sketch does not satisfy the differential privacy property. The intuition is as follows. Let's consider the following two databases:
>
> $D_1=[x_1,\cdots,x_1, x_1],$
>
> $D_2=[x_1,\cdots,x_1,x_2],$
>
> where both databases have $N$ items. If the adversary queries the frequency of $x_1$, under database $D_1$, the output will always be $N$. However, under database $D_2$, the query result for $x_1$ is highly unlikely to be $N$. For the query result of $x_1$ under $D_2$ to be $N$, the algorithm must hash $x_1$ and $x_2$ into the same index for at least half of the rows in the Linear Sketch, which occurs with very low probability. Therefore, the vanilla linear sketch does not satisfy DP except for very large $\epsilon$.
>
> W3: Thanks for raising this interesting question. For the Private Count-Min, we initialize the counter C with $E$ plus some noise sampled from a Gaussian distribution. The reason for adding $E$ to Counter C, is to preserve the property of no frequency underestimation with high probability. Even if the adversary knows about $E$, the differential privacy guarantee still holds as noise sampled from the Gaussian distribution is also added to Counter C. In addition, as suggested by other reviewers, we improved the uniform bound for our algorithm and provided detail comparison between our work and a concurrent work in the revision.

---

### Official Review · Reviewer_a3xv · 2022-07-25

**Rating:** 4
**Confidence:** 4
**Soundness:** 3 good
**Presentation:** 3 good
**Contribution:** 2 fair

**Summary:**

The paper shows that the Count (Median) Sketch and the Count Min sketch for frequency estimation can be made differentially private by adding a small amount of Gaussian noise. The sketches are then used for quantile estimation, similarly to how the non-private sketches are used for the same problem.

**Questions:**

I would suggest giving either an improved analysis or a proof that your analysis is tight for either sketch. Alternatively, a less immediate application of the private sketches can also strengthen your paper.

**Limitations:**

No comment.

**Strengths And Weaknesses:**

For better or worse, this is a very straightforward application of textbook differential privacy techniques. The small dimension of the sketch leads to a good bound on the ell_2 sensitivity, which allows adding a small amount of Gaussian noise to achieve privacy. This is a useful tool to have, but, in my opinion, a straightforward application of the Gaussian noise mechanism to a well-known sketch is not enough to meet the bar for publication.  The paper needs either a tighter analysis (note that a recent preprint by Pagh and Thorup claims a tighter bound on the error of the Count Sketch with added Gaussian noise), or a more impressive application.

---

> ### Author Response · Authors · 2022-07-31
> **Response to Reviewer a3xv**
>
> Thank you for your detailed responses, thoughtful suggestions, and bringing the new preprint to our attention. Indeed, the concurrent work [1] and our work share the identical algorithm for Differentially Private (DP) Count-Median. However, in addition to DP Count-Median, we also showcased DP Dyadic Count-Median, an important extension of DP Count-Median to approximate quantiles. To strengthen our contributions, we also discussed the continuous release of frequency estimations for the stream setting in the Appendix D of the revision.
>
> Moreover, we disagree with the claim that [1] has a better bound than our work. In the revision, we present an improved analysis of our algorithm, showcase that our algorithm provide the tight uniform bound (Appendix E), and add a new section to discuss related works, and include a detailed comparison (Appendix C) between our work and [1].
>
> In fact, there is a major difference between the analysis in [1] and ours. For the following analysis, let $f(x)$ denote the true frequency of item $x$, $\widetilde{f}(x)$ denote the non-private estimate of item $x$ (the output of the original Count-Median), and $\widehat{f}(x)$ denote the private estimate of item $x$ (the output of the Private Count-Median with the same set of hash functions). While [1] focused on the point-wise bound for $|\widehat{f}(x)-f(x)|$ , we focused on the uniform bound for $\sup_{x}|\widehat{f}(x)-\widetilde{f}(x)|$ (updated in the revision). We are interested in the trade-off between privacy and accuracy for the Count-Median sketch. In particular, we want to answer the question of what additional error will be imposed on the estimated frequency due to the Differential Privacy guarantee. As a result, our result shows a uniform bound $\sup_{x}|\widehat{f}(x)-\widetilde{f}(x)| \leq E$ where $E$ is a function of the desired accuracy, failure probability, and privacy guarantee and, shown in Appendix E, our algorithm's uniform bound is tight when universe is large (often the case in Big Data era). To derive the point-wise bound for $|\widehat{f}(x)-f(x)|$, we can simply combine our result with any point-wise bound for $|\widetilde{f}(x)-f(x)|$ due to the triangle inequality. In contrast, [1]'s analysis can not imply even point-wise bound for $|\widehat{f}(x)-\widetilde{f}(x)|$. In summary, for linear sketch with $k$ rows:
>
> Our uniform bound (Theorem 3.2): $\sup_x |\widehat{f}(x)-\widetilde{f}(x)|\leq E=\widetilde{O}(\sqrt{\frac{k}{\rho}})$.
>
> Our lower bound (Theorem E.1): $\sup_x |\widehat{f}(x)-\widetilde{f}(x)|\geq \Omega(\sqrt{\frac{k}{\rho}})$.
>
> The point-wise bound in [1] (reformulated for comparison.): $|\widehat{f}(x)-f(x)|\leq \text{non-private error bound} + \widetilde{O}(\sqrt{\frac{1}{\rho}})$.
>
> We agree that [1] has a tight point-wise bound for $|\widehat{f}(x)-f(x)|$ by using the concentration of median. By comparing our analysis and [1] for $|\widehat{f}(x)-f(x)|$, [1]'s point-wise bound removes the $\sqrt{\log\frac{2}{\beta}}$ ($\log\frac{2}{\beta}$ detes the number of rows in linear sketches) in the lower order term $E$ which is added to the original estimation error $|\widetilde{f}(x)-f(x)|$. However, the difference may not be substantial. When the database is large (which is the usual case for the need to perform approximations), $E$ is small compared to the $|\widetilde{f}(x)-f(x)|$. Even for the extreme case of $\beta = 1e-10$, the amplification factor for calculating $E$ is $\sqrt{\log\frac{2}{\beta}} \approx 5.8$, which $E$ is still very likely to be small compare to $|\widetilde{f}(x)-f(x)|$.
>
> We believe that neither of the two results dominates the other. Both [1]'s point-wise bound for $|\widehat{f}(x)-f(x)|$ and our uniform bound for $|\widehat{f}(x)-\widetilde{f}(x)|$ are useful analyses for understanding Differentially Private Count-Median with Gaussian noise.
>
> [1] Improved Utility Analysis of Private Count Sketch, Pagh et al.

---

> > ### Comment · Reviewer_a3xv · 2022-08-05
> > **Thank you for explaining the relationship to [1]**
> >
> > Thank you for this detailed explanation of the relationship between your results and those in [1]. I agree that the guarantees in the two papers are not necessarily comparable.
> >
> > My main concern is still that the paper’s main contribution is a straightforward application of a textbook differential privacy technique to a classical DP sketch.

---

> > > ### Author Response · Authors · 2022-08-06
> > > **Thank you for the feedback.**
> > >
> > > Thank you for reading our revision and rebuttal.
> > >
> > > We agree that DP is obtained via standard techniques, e.g., the Gaussian mechanism. We wouldn’t necessarily think this is a limitation. Simplicity is often greatly valued, and it is a *zen* to create algorithms using exclusively Gaussian mechanisms so one can get tight privacy accounting and composition from modern CDP/RDP/f-DP literature. Most highly impactful applied DP work is just about using standard techniques for solving practical problems (e.g., the seminal paper of Abadi et al. [1] did not invent the Sampled-Gaussian Mechanism or DP-SGD, but they applied these techniques successfully to deep learning). Our main contribution is to theoretically and practically demonstrate that strong DP guarantees can be achieved with high utility in linear sketching — which the reviewer agrees to be a classical and widely-used family of methods.
> > >
> > > Two specific suggestions were raised in the initial review: (1) showcase that our analysis is tight, and (2) provide a less immediate application. To incorporate the first suggestion, we improved our analysis and showcased that our uniform bound is tight. While the DP Count-Median proposed in [2] and ours are identical by applying the standard/text-book Gaussian Mechanism on the classic Count-Median, the analyses are different, and we believe both works contain valuable insights and give tight theoretical analysis.
> > >
> > > To incorporate the second suggestion, we extend our algorithms to the data stream setting. We plan to make the algorithms more comprehensive by adding a more detailed treatment to the data-stream setting. In particular, we plan to implement and adapt a recently proposed method by [3] (a more practical generalization of the binary tree mechanism described in Appendix D).
> > >
> > > In this work, we applied the standard Gaussian Mechanism to classical linear sketches. With tight theoretical analysis and extensive experiments, we demonstrated that the DP linear sketches with Gaussian Mechanism have a high utility privacy trade-off for summarizing many essential statistics such as frequency, top-k, and quantile approximations. We believe that the simplicity in our algorithms is valuable and enables our work to be more widely applicable to real-world applications and systems for protecting privacy while achieving accurate estimations.
> > >
> > > [1] Deep learning with differential privacy. Abadi et al.
> > >
> > > [2] Improved Utility Analysis of Private Count Sketch, Pagh et al.
> > >
> > > [3] Constant matters: Fine-grained Complexity of Differentially Private Continual Observation. Fichtenberger et al.

---

### Meta-Review · Area_Chair_C4iA · 2022-08-27

**Recommendation:** Accept
**Confidence:** Less certain

**Metareview:**

This work constructs differentially private linear sketches for frequency estimation and related problems. It shows that count min/median sketch can be made DP by adding noise. This is an interesting contribution.
The reviewers had concerns about the novelty of the paper. The (updated version of the) paper does a reasonable analysis of the algorithm. The authors point out in the rebuttal how their bounds are incomparable to a concurrent work. I am not convinced that the incomparable parts are interesting in the sense that closeness to the non-private count sketch does not seem like a useful goal in itself, and the target of interest is always the original frequencies.
The authors do an empirical analysis and show that the algorithm works well under reasonable privacy budgets. Overall, I think this paper may be a reasonable one to accept.

**Award:**

No

---

### Decision · Program_Chairs · 2022-09-14

Accept